# DEEP FUSION OF MULTI-ATTENTIVE LOCAL AND GLOBAL FEATURES WITH HIGHER EFFICIENCY FOR IMAGE RETRIEVAL

## ABSTRACT

Image retrieval is to search images similar to the given query image by extracting features. Previously, methods that firstly search by global features then re-rank images using local feature matching were proposed, which has an excellent performance on many datasets. However, their drawbacks are also obvious. For example, the local feature matching consumes time and space greatly, the re-ranking process weakens the influence of global features, and the local feature learning is not accurate enough and semantic enough because of the trivial design. In this work, we proposed a Unifying Global and Attention-based Local Features Retrieval method (referred to as UGALR), which is an end-to-end and single-stage pipeline. Particularly, UGALR benefits from two aspects: 1) it accelerates extraction speed and reduces memory consumption by removing the re-ranking process and learning local feature matching with convolutional neural networks instead of RANSAC algorithm; 2) it learns more accurate and semantic local information through combining spatial and channel attention with the aid of intermediate supervision. Experiments on Revisited Oxford and Paris datasets validate the effectiveness of our approach, and we achieved state-of-the-art performance compared to other popular methods. The codes will be available soon.

## 1 INTRODUCTION

Image retrieval is a classic task in computer vision. Its primary purpose is to search images that are similar to the given image through extracting features. Learning image representations well is the key factor to superior performance for image retrieval.

Global and local features are two typical image representations in image retrieval. According to (Cao et al., 2020), global features summarize image content and could learn similarity across very different poses, but lack spatial arrangement information. Local features contain geometric information and descriptors about specific regions, but lack global understanding of image content. For better performance, popular approaches first search with global features, then re-rank with local feature matching. This kind of method could achieve state-of-the-art results according to (Cao et al., 2020)(Taira et al., 2018)(Sarlin et al., 2019). Nevertheless, they have substantial drawbacks when put into practical use. First, the processes of extracting local features, local feature matching, and re-ranking are of low efficiency either in memory or in latency. For example, extracted local features often occupy too much memory. The RANSAC algorithm used in local feature matching is extremely time-consuming and also unstable since it depends heavily on iteration times. What's more, it needs to rank twice to get the final retrieval results and re-ranking with local features would erase the influence of global features in top results; Second, the local attention module is too trivial to capture local information good enough in both accuracy and semantics. Besides, previous works mostly treat attention and features equally (Ng et al., 2020)(Woo et al., 2018)(Hu et al., 2018), but rigit balance factor between attention and features would limit attention learning.

A question arises: facing the problem that retrieval results are correct in the category but lack attention to detail, is there any other way to solve it without the downsides mentioned above? Our work gives an appropriate answer. With a different view, we could interpret this problem as that the learned global features are not powerful enough to capture local information, while local fea-

tures perform better in distinguish details. Since local feature matching is to find the appropriate homography matrix between images, which is a process of learning linear transformation, we think it could be learned by the convolutional neural network. Based on these ideas, we intend to learn a more powerful global feature that contains merits of local features with the aid of multi-attentive local attention, CNN-based homography transformation, and information fusion.

First, we focus on local feature attention. We work on enhancing it in both module design and training strategy. In terms of module design, we proposed a LALM (Location Attention Learning Module) block that contains both spatial attention mechanism and channel attention mechanism to enhance semantics and accuracy. In terms of training strategy, we introduced intermediate supervision into the process of training attention modules with the aim of giving different importance to attention and features.

Second, we focus on learning CNN-based homography transformation and information fusion. We concatenate the output of LALM with the global feature branch, let subsequent blocks learn homography transformation in local feature matching, and naturally fuse it with global features. Instead of extracting global and local features respectively from the trained model, we only use the enhanced features that contain both global and local information to perform image retrieval, which reduces time and space costs caused by local features storing, matching, and re-ranking.

In general, our main contributions are summarized as follows:

- We innovatively utilised CNN network to learn the homography transformation in local feature matching and generate a more powerful feature that contains both global information and local information.
- We proposed a LALM(Local Attention Learning Module) block which learns local attention in both spatial and channel dimension. We introduced intermediate supervision into image retrieval to help the network treat attention and feature differently and adaptively, which effectively avoids attention degradation.
- We proposed a single-stage model named UGALR(Unify Global and Attention-based Local Features Retrieval methods) that can be trained end-to-end. It achieved state-of-the-art performance on two typical datasets with less memory and faster extraction speed compared with other popular methods.

## 2 RELATED WORKS

### 2.1 IMAGE DESCRIPTORS

Image descriptors refers to feature vectors that contain image content information, including local descriptors and global descriptors. Local descriptors generally contain both key points location and corresponding features, and some popular methods include (Noh et al., 2017), (Tolias et al., 2016a), (Gong et al., 2014) and (Arandjelovic et al., 2016). The global descriptor is in the form of a compact vector, which is usually generated by a single single forward-pass of CNN. And some typical approaches include(Ng et al., 2020), (Radenovic et al., 2019),(Deng et al., 2019),(Sohn, 2016). Local features tend to focus on local information but have poor performance on the global, while global features summarize the content of an image but lose geometry information in it. Consequently, more and more cases began to combine the two kinds of features, such as (Sarlin et al., 2019), (Cao et al., 2020), (Taira et al., 2018), and (Siméoni et al., 2019). However, this kind of approach also suffers from great time and space consumption. In this work, we propose a new idea that learns homogaphy information with CNN to avoid huge time and space consumption.

### 2.2 HOMOGRAPHY ESTIMATION

Homography estimation is to find transformation matrix between images, which are taken from different perspectives of the same object. In traditional approaches, solving homography often needs to detect and match features, then using estimation algorithm, such as (Fischler & Bolles, 1981), (Chen et al., 2017), and (Barath et al., 2019), to establish correspondences between images. But these approaches suffer from great time and space consumption. With the development of CNN, more deep homography approaches were proposed, which include supervised (DeTone et al., 2016)

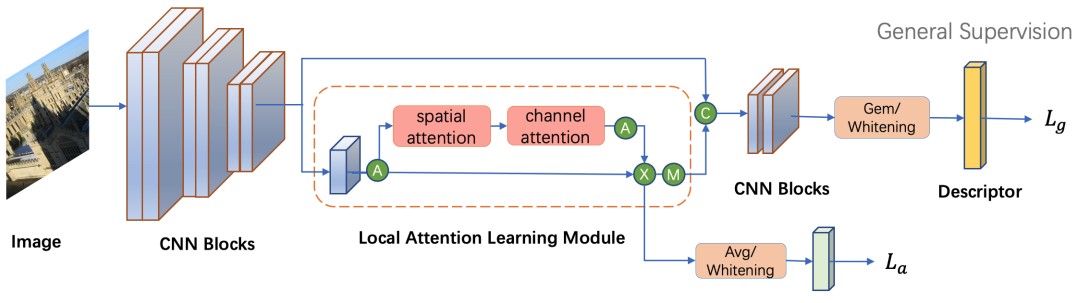

Figure 1: Illustration of UGALR. Firstly, the images are fed into CNN blocks to extract features. Then the output features are fed into the Local Attention Learning Module, which is denoted with the orange dotted box, to learn attention-based local features. There are two small branches in LALM. One branch is to learn attention score for feature maps, the other branch is the original feature maps without any operation. The LALM output is the multiplication result of them, which is attention-weighted feature maps and contains local information. The weighted feature maps, which then go through avg-pooling and whitening operation, are intermediately supervised by $La$. On the other hand, the learned weighted feature maps goes through avg-pooling in the channel dimension and are then concatenated with originally extracted features to feed into the subsequent CNN blocks, Gem-pooling, and whitening FC-layers, which is supervised by $Lg$. "X", "C", "A", and "M" respectively denote element-wise multiplication, concatenation, activation functions and average pooling in the channel dimension.

and unsupervised approaches (Nguyen et al., 2018). However, these methods are not suitable for image retrieval.

## 2.3 INTERMEDIATE SUPERVISION

Intermediate Supervision is a training technique that provides supervision in the intermediate part of the model. This technique is previously used in landmark detection (Newell et al., 2016) (Wei et al., 2016) (Carreira et al., 2016) (Pfister et al., 2015) because of the ability of emphasizing critical features and easing gradient vanishing. Nevertheless, to the best of our knowledge, there is no case that combines intermediate supervision with image retrieval yet.

## 3 METHODOLOGY

In this section, we first present our design motivation, then introduce model construction, followed by the retrieval process.

### 3.1 MOTIVATION

As illustrated in the introduction, to learn better local attention and address the time consuming, memory consuming and low efficiency problems of using local features, we intend to integrate multi-attentive local information with the learning of global features and let CNN learn homography transformation in local feature matching. Our idea is to learn more powerful global features that could distinguish details in images to get the same or superior retrieval results with faster extraction speed and less memory occupation compared with the global and local re-raking methods.

The implementation and theories of UGALR will be described in detail in the next section.

### 3.2 MODEL

Our proposed UGLAR consists of a backbone network and a local attention module named LALM(Local Attention Learning module), as shown in Figure 1. We first introduce the LALM block, then illustrate the backbone network from a global perspective.

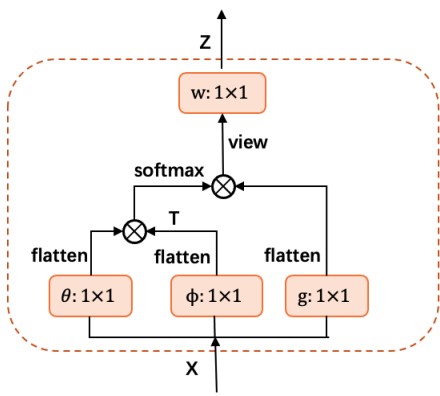

Figure 2: Illustration of spatial attention module. Here we take the relationship learning part of non-local block(Wang et al., 2018)as the spatial attention module. X is the input of the module and Z is the output. g, $\theta$, $\phi$ and w denote 1x1 convolution kernels.

**Local Attention Learning Module** As discussed above, LALM is proposed to learn multi-attentive local attention and features. In this block, the input tensor firstly goes through a convolution kernel followed by ReLU to reduce dimensions, then goes through a spatial attention module, which is a modified non-local block(Wang et al., 2018), then goes through a channel attention module, which is a $1 \times 1$ convolution kernel, and then activated by Softplus(Dugas et al., 2000).

Since the non-local block has the ability of learning relationships among features in the spatial dimension of a 3d feature map tensor, we could interpret it as a kind of spatial attention. Considering the function of spatial weighting and the requirement of low latency, we modified the non-local block and delete the residual connection in it, as shown in Figure 2. Formally, let X denote the input of the spatial attention module, g, $\theta$, $\phi$, and w denote 1x1 convolution kernels. The whole calculation process could be expressed as the formulas below.

$$\theta_v = flatten(\theta(X)), \phi_v = flatten(\phi(X)), g_v = flatten(g(X)) \tag{1}$$

$$X^{'} = softmax(\theta_v \times \phi_v^T) \tag{2}$$

$$Y = view(X^{'} \times g_v) \tag{3}$$

$$Z = w(Y) \tag{4}$$

where $\times$ represents matrix product, $\theta(\cdot)$, $\phi(\cdot)$, $g(\cdot)$, $w(\cdot)$ are convolutional calculation, and $flatten$, $softmax$, $view$ are their corresponding operations.

The channel attention module contains a 1x1 convolution kernel. Since it and the spatial attention module's output have the same number of channels, after training, each of these parameters in the module corresponds to a weight of one channel in feature maps. Thus we could interpret it as a process of learning channel attention. The reason why we do not use other types of channel attention, such as SE block(Hu et al., 2020), is to reduce extraction latency.

To learn the most optimal attention for feature maps, we not only need to know which point in the spatial dimension is critical but also need to figure out which characteristics of the feature map significantly make sense. Thus, the combination of modified non-local blocks and 1x1 convolution kernel here plays an important role in attention learning. Softplus function maps the output of attention modules into a positive feature map-$G$. And it represents the attention score. We then element-wise multiply $G$ and $X$ to get $D$, which is a weighted feature map that contains information of key local attention.

Intermediate supervision is added to the LALM block. As shown in Figure 1, $D$ is then followed by avg-pooing and whitening FC-layer, which is supervised with cross-entropy loss.The cross-entropy loss can be expressed as:

$$L_a = -\frac{1}{N}\sum_{i=1}^{N} log \frac{e^{W_{y_i}^T x_i + b_{y_i}}}{\sum_{j=1}^{n} e^{W_j^T x_i + b_j}} \tag{5}$$

where $x_i \in \mathbb{R}^d$ represents the feature of the i-th image with label $y_i$. d is the dimension of embedding features. N denotes the batch size and n is the number of classes. $b_j \in \mathbb{R}^n$ denotes the bias term and $W_j \in \mathbb{R}^d$ is the j-th column of the weight $W \in \mathbb{R}^{d \times n}$.

Intermediate supervision is indispensable since it differentiates the importance of attention and features. Otherwise, rigid factor between attention and features would limit attention learning, which leads to inaccuracy and fuzzy semantics caused by attention degradation. With intermediate supervision, the network could learn attention and features differently and adaptively based on specific tasks. Furthermore, the addition of it alleviates vanishing gradient and model degradation caused by "dying ReLU", making the learned attention more accurate.

In addition, $D$ goes through average pooling in the channel dimension to produce $O$, and $O$ is subsequently concatenated with the middle block of backbone networks and the result of concatenation is put into the next block. Taking in a $C \times H \times W$-shaped feature map, average pooling in the channel dimension outputs a feature map with the shape of $1 \times H \times W$, which is of great significance. As illustrated above, the backbone network leads the training process intending to learn more powerful features, while LALM plays an auxiliary role to learn more accurate and semantical local information. If we crudely concatenate the output of LALM with the global branch, the backbone network's dominant position would be weakened, since over-contribution of LALM leads to intermediate supervision interfering with global features training, which results in suboptimal performance. Average pooling in channel dimension not only reduces the dimension of feature maps to avoid this problem but also preserves key points information.

**Backbone Network** Given an input image, we apply a convolutional neural network as backbone and take the block i output $X_i$ shaped of $C_i \times H_i \times W_i$. $C_i$, $H_i$, $W_i$ here correspond to the number of channels, height, and width respectively. With $X_i$ being the input, we plugged LALM into this block. Then we would get a weighted and dimension-reduced feature map named $O_i$ that contains local information. The concatenation of $O_i$ and $X_i$ is then fed into the next block of backbone.

The innovative design of concatenation and feeding into next blocks has two merits. Firstly, the addition of $O_i$ provides extra attention to boost global features learning. Secondly, the subsequent blocks could utilise the key attention information to learn homography transformation in local feature matching. We assume that there is an identical average target available for each category of objects. Then, from each image in one category to its corresponding target image exists a homography matrix. Our purpose is to let CNN learn the transformation and fuse it with global features.

In general, there is a pooling operation at the end of the backbone to integrate important features in feature maps. Here we choose generalized mean pooling(GEM)(Radenovic et al., 2019), because it combines the advantages of max-pooling(Tolias et al., 2016b) and avg-pooling(Babenko & Lempitsky, 2015) and its effectiveness has been widely proved. After GEM pooling, we use a fully connected layer to whiten the aggregated features, as illustrated in (Gordo et al., 2017b).

In terms of general supervision loss, we need to learn global features with strong discrimination ability, which are compact within classes and distant between classes, so we adopt the ArcFace loss(Deng et al., 2019) here. As an additive angular margin loss, it outperforms most losses in metric learning under the same settings, as illustrated in (Srivastava et al., 2019). And the Arcface loss can be expressed as:

$$L_g = -\frac{1}{N} \sum_{i=1}^{N} log \frac{e^{s(cos(\theta_{y_i}+m))}}{e^{s(cos(\theta_{y_i})+m)} + \sum_{j=1,j!=i}^{n} e^{scos\theta_j}} \qquad (6)$$

where N is the batch size, n is the number of classes. $\theta_j$ denotes the angle between the weight $W_j$ and the feature $x_i$, where $x_i \in \mathbb{R}^d$ is the L2-normalized feature of the i-th image with $||x_i|| = 1$, $W_j$ is the j-th column of the L2-normalized weight $W \in \mathbb{R}^{d \times n}$ with $||W_j|| = 1$. d is the dimension of embedding features. $y_i$ represents the label of $x_i$. m is an additive angular margin penalty between $x_i$ and $W_{y_i}$ and s is a re-scale factor.

The total loss is the weighted sum of intermediate supervision loss and general supervision loss, which can be expressed as:

$$L = L_g + \lambda L_a \qquad (7)$$

where $\lambda$ is a weight factor between $L_g$ and $L_a$.

### 3.3 RETRIEVAL

There are two steps in the retrieval stage: extracting image features and measuring similarity between features. Previous works(Gordo et al., 2017a)(Noh et al., 2017) (Radenovic et al., 2017)(Cao et al., 2020) have shown us that multi-scale information is of great significance in feature extraction. In order to introduce multi-scale information, we use 5 scales, $[s_1,s_2,s_3,s_4,s_5]$, to resize the original image and we could obtain five global features of different scales for one image. These features are then L2-normalized, summed up and L2-normalized again to get the final embedding $F$. Besides, we use dot product of two features to measure similarity. Since global features are L2-normalized, the dot product denotes cosine distance between two features.

## 4 EXPERIMENTS

### 4.1 DATASETS

**Training datasets** In this work, the clean version of Google Landmark V2(Ozaki & Yokoo, 2019), which contains 1580470 images and 81313 classes, serves as a training dataset. We split 80% of images into the training set and 20% of images into the eval set.

**Test datasets** Revisited Oxford and Paris(Radenovi et al., 2018) serve as test datasets, which are denoted as $\mathcal{R}$Oxf5k and $\mathcal{R}$Par6k. $\mathcal{R}$Oxf5k contains 4993 database images and $\mathcal{R}$Par6k contains 6322 database images. They respectively have 70 query images. We also use $\mathcal{R}$1M distractor set with 1 M images to evaluate model performance. By convention, the mean average precision(mAP) is reported as performance metrics.

### 4.2 SETTINGS

In this section, we will introduce model setup, training details and extraction settings in turn.

**Model setup** In this work, ResNet-50 serves as the backbone. We denote the *block3* output as $X$, which is shaped of $1024 \times H \times W$. LALM is fed with $X$ and outputs feature map O shaped of $1 \times H \times W$. The input of *block4* of the backbone network is shaped of $1025 \times H \times W$. The parameter p of GeM pooling is set as 3. The whitening FC-layer reduces the dimension of features from 2048 to 512. In Arcface loss, we set the margin m=0.15 and set the scale=30. In terms of the total loss L, the $\lambda$ is set as 0.6.

**Training details** Our model is implemented using PyTorch. Data augmentation methods include random flip, random crop, and resizing images into $512 \times 512$. The weights of models are initialized from pre-trained ImageNet. We used a batch size of 128 and applied some learning rate adjustment strategies: cosine decay and warm up. The initial learning rate is 0.05. After 100 epochs using the SGD optimizer, we take the best performing model and report its results on different datasets.

**Extraction settings** With the extraction and retrieval process introduced in section 3.3, we illustrate some parameter settings here. The multiple scales we use to resize images include 0.3535, 0.5, 0.7071, 1.0, and 1.4142. The final extracted features $F$ are shaped of $1 \times 512$. $F$ is directly used to perform retrieval and no more re-ranking process is needed.

### 4.3 COMPARSION WITH STATE-OF-ART METHODS

In this part, we compare our approach with other state-of-the-art techniques in two dimensions. One is accuracy, the other is memory and extraction speed. **Accuracy** Tab.1. shows the comparison between our methods and the retrieval state-of-the-art in accuracy. The table includes three parts. The first part includes local features aggregation-based methods. The second part includes global features/global+local features-based methods. Notice that those marked as ("global+local") denote approaches that firstly search with global features then re-rank with local feature matching, and those marked as "(Re)" are implemented under the same settings as ours. And the last part is our approach. As we can see, UGALR completely outperforms previous global features based methods, for $\mathcal{R}$oxford5k, $\mathcal{R}$paris6k and their $\mathcal{R}$1M distractor sets. The largest two improvements we made are 7.67% in $\mathcal{R}$Oxf-H, and 9.31% in $\mathcal{R}$Par-H. In addition, our approach even surpasses the stats-of-art global and local features-based method, such as DELG(Cao et al., 2020), on most

Table 1: **Comparison to state-of-the-art retrieval methods** We present MAP(mean average precision) of different approaches on ROxf5k/RPar6k and ROxf5k+1M/RPar6k+1M datasets in both Medium and Hard protocols.

| Methods | Medium | | | | Hard | | | |
|---|---|---|---|---|---|---|---|---|
| | $\mathcal{R}$Oxf | +1M | $\mathcal{R}$Par | +1M | $\mathcal{R}$Oxf | +1M | $\mathcal{R}$Par | +1M |
| **(I) Local features** | | | | | | | | |
| HesAff-HardNet-ASMK*+SP | 65.60 | - | 65.20 | - | 41.10 | - | 38.50 | - |
| DELF-ASMK*+SP | 67.80 | 53.80 | 76.90 | 57.30 | 43.10 | 31.20 | 55.40 | 26.40 |
| **(II) Gocal features** | | | | | | | | |
| **/ Global+Local features re-ranking** | | | | | | | | |
| AlexNet-GeM | 43.30 | 24.20 | 58.00 | 29.90 | 17.70 | 9.40 | 29.70 | 8.40 |
| VGG16-GeM | 61.90 | 42.60 | 69.30 | 45.40 | 33.70 | 19.00 | 44.30 | 19.10 |
| R101-R-MAC | 60.90 | 39.30 | 78.90 | 54.80 | 32.40 | 12.50 | 59.40 | 28.00 |
| R101-GeM | 64.70 | 45.20 | 77.20 | 52.30 | 38.50 | 19.90 | 56.30 | 24.70 |
| R101-SOLAR | 69.90 | 53.50 | 81.60 | 59.20 | 47.90 | 29.90 | 64.50 | 33.40 |
| R50-DELG | 73.60 | 60.60 | 85.70 | 68.60 | 51.00 | 32.70 | 71.50 | 44.40 |
| R50-DELG(Re) | 76.07 | 61.51 | 85.48 | 68.42 | 49.91 | 33.10 | 70.60 | 43.93 |
| R50-DELG(global+local) | 78.30 | - | 85.70 | - | 57.90 | - | 71.00 | - |
| R50-DELG(global+local)(Re) | 79.02 | - | 88.90 | - | **58.00** | - | 76.18 | - |
| **(III) Ours** | | | | | | | | |
| R50-UGALR | **79.98** | **66.68** | **90.85** | **70.48** | 57.58 | **36.81** | **79.91** | **46.95** |

protocols. For example, on rparis6k dataset: $1.95\%$ improvement was made in $\mathcal{R}$Par-Medium and $3.73\%$ improvement was made in $\mathcal{R}$Par-Hard. On roxford dataset, we made an improvement of $0.96\%$ in $\mathcal{R}$Oxf-Medium. Surprisingly, our approach using only global features outperforms the state-of-art global feature+local feature re-ranking methods.

Overall, our approach achieved state-of-the-art performance in accuracy compared with the other advanced methods. These results validate the effectiveness of our ideas: combining channel mechanism, spatial mechanism, and CNN-based homography transformation greatly enhanced the performance of global features, which have more ability to distinguish details.

Table 2: **Experiments for extraction speed and memory of different approaches**. We did this experiment on the same NVIDIA Tesla P40 GPU, with images resized into 1024. $UGALR^*$ refers to a version of $UGALR$ with the same extraction settings with Group 3.

| Methods | Extraction | Memory | |
|---|---|---|---|
| | latency(ms) | $\mathcal{R}$Oxf+1M | $\mathcal{R}$Par+1M |
| **(1) Local features** | | | |
| DELF-R-ASMK* | 2260 | 27.6 | - |
| DELF(3scales) | 98 | 434.2 | 434.8 |
| DELF(7 scales) | 201 | 477.9 | 478.5 |
| **(2) Global+Local features re-ranking** | | | |
| R50-DELG | 211 | 485.5 | 486.2 |
| R50-DELG(3 scales global+local) | 118 | 439.4 | 440.0 |
| R101-DELG | 383 | 485.9 | 486.6 |
| R101-DELG(3 scales global+local) | 193 | 437.1 | 437.8 |
| **(3) Global features** | | | |
| R50-GeM | 100 | 7.7 | 7.7 |
| R101-GeM | 175 | 7.7 | 7.7 |
| **(4) Ours** | | | |
| R50-UGALR* | 99 | 7.8 | 7.8 |
| R50-UGALR | **68.7** | **3.9** | **3.9** |
| R101-UGALR* | 177 | 7.8 | 7.8 |
| R101-UGALR | **132** | **3.9** | **3.9** |

**Speed and Memory**

Besides accuracy, speed and memory are also very important indicators to measure the model's performance. Tab.2 compares the extraction latency and feature memory of UGALR with the state-

of-the-art methods. In extraction latency, R50-UGALR runs with 68.69(ms) and R101-UGALR runs with 132(ms). In memory, features from R50-UGALR and R101-UGALR both occupy 3.9M, which are superior to other approaches. Methods marked with "*" used the same extraction settings as Group 3, which didn't reduce dimensions. As shown in Tab.2, when not reducing dimensions, UGALR outperforms local features based and unifying global and local features based methods in both extraction speed and memory, which performs similarly with global features based approaches. Dimensionality reduction through the FC-whitening layer further boosts its performance in its memory occupation and extraction latency, which are more suitable for practical use.

## 4.4 ABLATION STUDY

In this section, we performed an ablation study on different innovations of UGALR.

**Intermediate supervision** Tab.3. compares the effects of using and not using intermediate supervision. The first row presents the result of not using intermediate supervision and the second row presents the other. Experiments show that adding supervision in the intermediate part of UGALR brings an improvement of 1.52% in $\mathcal{R}$Oxf-M, 3.57% in $\mathcal{R}$Oxf-H, 1.75% in $\mathcal{R}$Par-M and 3.22% in $\mathcal{R}$Par-H, which is in line with our expectations, since intermediate supervision differentiates the importance of attention and features to avoid the degradation of attention learning.

Table 3: **Experiments for Intermediate Supervision**.

| Intermediate supervision | roxford5k | | rparis6k | |
|---|---|---|---|---|
| | M | H | M | H |
| no | 78.46 | 54.01 | 89.1 | 76.69 |
| yes | **79.98** | **57.58** | **90.85** | **79.91** |

**Local Attention Learning Module** Tab.4. reports the effects of LALM and homography transformation. The first row denotes the performance of not using the LALM block and the second row denotes the performance of using LALM. The model with LALM outperforms the model without it by 1.77% in $\mathcal{R}$Oxf-M, 1.27% in $\mathcal{R}$Oxf-H, 1.85% in $\mathcal{R}$Par-M and 3.74% in $\mathcal{R}$Par-H, which verified the effectiveness of the Local Attention Learning Module. As we can see, the module helps learn more accurate local attention and its concatenation allows the global branch to learn homography transformation and further boosts global features learning.

Table 4: **Experiments for Local Attention Learning Module**.

| LALM | roxford5k | | rparis6k | |
|---|---|---|---|---|
| | M | H | M | H |
| no | 78.21 | 56.31 | 89 | 76.17 |
| yes | **79.98** | **57.58** | **90.85** | **79.91** |

Table 5: **Experiments for exploring different kinds of attentions**

| Attenion setup | roxford5k | | rparis6k | |
|---|---|---|---|---|
| | M | H | M | H |
| no | 78.53 | 54.31 | 86.99 | 72.29 |
| spatial | 78.6 | 55 | 89.96 | 78.48 |
| channel | 78.33 | 55.93 | 88.88 | 76.72 |
| spatial+channel | **79.98** | **57.58** | **90.85** | **79.91** |

**Local attention setup** Tab.5. compares the performance of no attention and three different kinds of attention in LALM. For example, the first row represents not using any attention, and the second row means using only spatial attention. Experiments show that combining spatial and channel attention mechanism outperforms other three approaches by a large margin for all four protocols. The spatial attention module helps identify the key points and the channel attention module helps identify the critical characteristics. The joint learning of them is essential to the LALM block.

Table 6: **Experiments for average pooling in channel dimension**

| Avg pooling | roxford5k | | rparis6k | |
|---|---|---|---|---|
| | M | H | M | H |
| no | 76.31 | 50.89 | 87.79 | 74.16 |
| yes | **79.98** | **57.58** | **90.85** | **79.91** |

**The most suitable block for LALM applying** To explore which part of the network we should add the LALM block to, we also conducted a comparative experiment as shown in Tab.7. This experiment is based on the ResNet-50 backbone. From the table, the setup of adding LALM to the block3 has a MAP of 79.98% in $\mathcal{R}$Oxf-M, 57.58% in $\mathcal{R}$Oxf-H, 90.85% in $\mathcal{R}$Par-M and 79.91% $\mathcal{R}$Oxf-M, which surpasses the other three setups. Since the shallower feature maps of the network are less semantical for attention learning, plugging LALM into block2 has an inferior performance. The reason why the other two setups are also suboptimal is that the deeper feature map from the block4 output is of relatively small size and it is less localizable to learn great local attention.

Table 7: **Experiments for most suitable blocks for LALM applying**

| block | roxford5k | | rparis6k | |
|---|---|---|---|---|
| | M | H | M | H |
| 2 | 77.44 | 52.17 | 88.55 | 75.81 |
| 3 | **79.98** | **57.58** | **90.85** | **79.91** |
| 4 | 78.1 | 55.43 | 90.32 | 79 |
| 3+4 | 78.8 | 54.52 | 89.16 | 76.69 |

**Average pooling in channel dimension** The significance of average pooling in the channel dimension is illustrated in Section 3.2, and we conduct an experiment to verify its necessity. As shown in Tab.6, the model with Avg-pooling in the channel dimension outperforms the one without it by 3.67% in $\mathcal{R}$Oxf-M, 6.69% in $\mathcal{R}$Oxf-H, 3.06% in $\mathcal{R}$Par-M and 5.75% $\mathcal{R}$Oxf-M. By reducing dimension in an appropriate way, it effectively avoids intermediate supervision's over interference to global feature learning and well preserves key information at the same time.

**Feature fusion** To find out the best way to fuse LALM output and the block output of network backbones, we also carried out an experiment. As shown in Tab.8, the first row represents using addition to fuse features, the second row means using multiplication to fuse features, and the third row means performing concatenation. It is noticed that performance of the third group surpasses the first two groups by a large margin. This is expected. Compared with simple addition and multiplication, concatenation avoids information loss and preserves complete local attention information to learn homography transformation in images.

## 5    CONCLUSIONS

We proposed a single-stage image retrieval pipeline named UGALR. It utilizes spatial and channel attention to learn key local information, then feeds it to CNN to learn homography transformation in images. Through information fusion, UGALR produces multi-attentive features that contain both global and local information. In this work, we learn homography transformation with CNN then replace the re-ranking process with information fusion to obtain more powerful features and overcome the low efficiency of local features in storage and matching. We also combine spatial and channel attention to improve accuracy and semantics in local attention, apply intermediate supervision to avoid the degradation of attention learning. The effectiveness of UGLAR has been validated with comprehensive experiments, which achieves state-of-the-art performance with lower memory occupation and faster extraction speed.

Table 8: **Experiments for feature fusion**

| fusion approach | roxford5k | | rparis6k | |
|---|---|---|---|---|
| | M | H | M | H |
| add | 74.41 | 48.87 | 87.82 | 74.21 |
| multiply | 77.87 | 54.07 | 89.53 | 77.4 |
| concatenate | **79.98** | **57.58** | **90.85** | **79.91** |

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
