# OpenReview forum: "Deep Fusion of Multi-attentive Local and Global Features with Higher Efficiency for Image Retrieval"
_ICLR.cc/2022/Conference — ICLR 2022 Submitted_

### Official Review · Reviewer_wJ79 · 2021-10-28

**Correctness:** 2
**Technical Novelty And Significance:** 2
**Empirical Novelty And Significance:** 2
**Recommendation:** 1
**Confidence:** 5

**Main Review:**

Strengths:
S1) Combining local and global image features is a promising research direction.
S2) The paper provides several experiments, with many ablations.
S3) Some performance improvements seem to be observed compared to previous work.

Weaknesses:
W1) The authors naively claim several times in the paper that their method is learning homography transformations (Introduction, Methodology, Experiments, Conclusions). However, one can clearly see that this is not the case. The authors are simply concatenating local feature maps with the ones from the global feature branch and supervising by ArcFace classification loss. The authors seem to be assuming that homography learning is implicitly happening, which is naive. Similarly, the abstract claims that the proposed method is “learning local feature matching with convolutional neural networks instead of RANSAC algorithm” – this is incorrect.
W2) The paper is not well written, and several parts are hard to understand. For example, in Fig 1 the paper has blocks named A, X, M, C – what do they refer to? Similarly, in page 4 the text start referring to “D” (“D is then followed (...)”), but D had never been introduced before. D (and O, which is also later introduced) should be added to Fig 1, and A/X/M/C should be explained in the caption/text. It is also very difficult to understand the way that local and global feature maps are combined – the paragraph “Backbone Network” and the one before it are quite confusing.
W3) The authors seem to have suspiciously excluded results from previous work. For example, Tab 1 does not show numbers for “R50-DELG (global+local)” in the “+1M” dataset setups, which are available in the exact same paper where the other numbers from the table were taken from. I suspect that they were not included because in half of the cases the proposed method underperforms compared to “R50-DELG (global+local)” – specifically, the ROxf+1M cases.
W4) In Tab 2, the authors simply take latency numbers reported in previous work and compare to their own for the new method, using a different GPU. The comparison does not seem appropriate.


**Summary Of The Paper:**

The paper proposes a new method to combine global and local image features, targeted at image retrieval applications. It designs a new local feature model branch where both spatial and channel attention are used. The local feature branch undergoes supervision directly (in the paper called “intermediate supervision”), and this branch’s output is also concatenated to the global feature branch’s output in order to produce a final image embedding at the end.

**Summary Of The Review:**

The paper addresses an important technical direction of fusing local and global image features. However, it has several major flaws, such as: claiming learnable matching/homography when none of that is happening, suspiciously excluding previous work’s performance numbers in some cases; poor writing.

---

> ### Author Response · Authors · 2021-11-19
> **Response to Reviewer wJ79**
>
> &nbsp;&nbsp;&nbsp;&nbsp;Thanks for your comments about our papers，and we will address some concerns below.
>
> Q1. “The authors naively claim several times in the paper that their method is learning homography transformations (Introduction, Methodology, Experiments, Conclusions). However, one can clearly see that this is not the case. The authors are simply concatenating local feature maps with the ones from the global feature branch and supervising by ArcFace classification loss. The authors seem to be assuming that homography learning is implicitly happening, which is naive. Similarly, the abstract claims that the proposed method is “learning local feature matching with convolutional neural networks instead of RANSAC algorithm” – this is incorrect.”
>
> A1: Maybe there are some misunderstandings here. It is not just simply concatenation, but with the aid of intermediate supervision to learn local key point information and provide this kind of information to global branch. We will illustrate the process below. Firstly,  we enhance local attention learning in two dimensions: spatial and channel, to learn more accurate and semantic attention information. Second, we apply intermediate supervision in training local attention modules to adaptively learn the local key point information about category responses((attention is to find key point response and  softplus activation act as a threshold filter to extract local key point information)). And intermediate supervision helps avoid attention degradation. Third, we provide the output of local attention module to the blocks on the backbone, and the global branch takes into the local key point information to  learn  homography transformation to boost global feature learning. Because of the existence of softplus activation and intermediate supervision, it is not a simple concatenation.
>
> &nbsp;&nbsp;&nbsp;&nbsp;In addition, we would illustrate the process of implicitly homography learning. The reason why the local feature is better at precision is that it contains geometric information. Based on geometric information of two images, we could use RANSAC algorithm to estimate homography matrix and use homography matrix to compute the inliers between two images. The inliers are the key to local features re-ranking. That means the core of local feature matching is to find appropriate homography matrices, which is a kind of linear transformation. It is known to all that linear transformation could be learnt by CNN. Thus we could use CNN to learn homography transformation. Our interpretion about learning homography transformation with CNN is as follow.
>
> &nbsp;&nbsp;&nbsp;&nbsp;In instance-level image retrieval, images in one category describe the same object. So we could assume that there is an identical average target available for each category of objects. Then, from each image in one category to its corresponding target image exists a homography matrix. For example, if we take the front image of the object as the standard target, then each image of this category could be transformed to the standard target using a homography matrix. The mission of our network is to learn homography transformation between them.
>
> &nbsp;&nbsp;&nbsp;&nbsp;Just like CNN could implicitly learn mappings from a raw image to its corresponding category, CNN could also implicitly learn the homography transformation from one image to its corresponding standard target of its category with local key point information provided. In fact, experiments have validated the effectiveness of our idea, and the network could learn local feature matching without the aid of RANSAC algorithm.
>
> &nbsp;&nbsp;&nbsp;&nbsp;Thanks for your careful reading and patience.

---

> > ### Comment · Reviewer_wJ79 · 2021-11-25
> > **Response to author's Q1 rebuttal**
> >
> > I read the author's response to this question in detail. My opinion did not change at all: the author's explanation is clearly handwavy. Just because the concatenation of local features with global ones works does not mean at all that some geometric transformation is learned. More likely, it's just adding some more localized information that is globally discriminative -- which is why performance is improved. There is no correspondence learning going on, there is no homography learning going on.

---

> ### Author Response · Authors · 2021-11-19
> **Response to Reviewer wJ79**
>
> Q2. "The paper is not well written, and several parts are hard to understand. For example, in Fig 1 the paper has blocks named A, X, M, C – what do they refer to? Similarly, in page 4 the text start referring to “D” (“D is then followed (...)”), but D had never been introduced before. D (and O, which is also later introduced) should be added to Fig 1, and A/X/M/C should be explained in the caption/text. It is also very difficult to understand the way that local and global feature maps are combined – the paragraph “Backbone Network” and the one before it are quite confusing."
>
> A2: Thanks for your valuable comments. Since the first version of this paper is more than 9 pages long, in the process of reducing length, we made some errors and accidentally missed some important content. And we should apologize for that. In the latest draft, we have fixed the errors you mentioned. Thanks very much.
>
> Q3. "The authors seem to have suspiciously excluded results from previous work. For example, Tab 1 does not show numbers for “R50-DELG (global+local)” in the “+1M” dataset setups, which are available in the exact same paper where the other numbers from the table were taken from. I suspect that they were not included because in half of the cases the proposed method underperforms compared to “R50-DELG (global+local)” – specifically, the ROxf+1M cases"
>
> A3: Notice that our reproducing process is based on the source code published by the authors of DELG on github, and the reason why we do not show numbers for "R50-DELG(global+local)" in the "+1M" dataset setups is that we can't reproduce the reported result  and we have to doubt the statistics in the "+1M" dataset setups in DELG. When reproducing on “+1M” dataset, we encountered great difficulties. R1M distractor set contains over 1000,000 images, and we need to extract local features for all of these images. The local features of one image comprise 1000 keypoint locations and 1000 keypoint features. Firstly we need to match features, then we use RANSAC algorithm to match locations. Just the stage of extracting local features takes 3 days. Just re-ranking local features of 30 images takes half an hour. If we re-rank all the local features of these images, it would take over 16,000 hours, which means 695 days. It's completely unbelievable since it takes too much time and occupies too much memory, which makes reproducing the process impossible. Consequently, we have to double the statistics of the reported results in the previous work. Our reproducing process is based on the source code published by the authors of DELG on github and we checked it again and again.  It is really weird.
>
> &nbsp;&nbsp;&nbsp;&nbsp;There is no need for us to deliberately miss the statistics, because even the numbers for “R50-DELG (global+local)” in the “+1M” dataset setups in the original paper is lower than the experiments results of our UGALR, and you can find it in the DELG paper. If we deliberately missed the statistics, why didn't we simply make up fake data to fill Table 1, which looks more "beutiful" and "perfect", and why did we choose to miss the statistics?  Moreover, UGALR focuses on reducing memory and extraction latency. It doesn't matter that its performance in “R50-DELG (global+local)” in the “+1M” dataset setups is lower than DELG, because its performance in “R50-DELG (global+local)” in the ROxf dataset setups has outperformed DELG. It's enough to prove that our method could achieve state-of-the-art performance with less memory and less time. So we have no reason to deliberately miss the statistics.
>
> &nbsp;&nbsp;&nbsp;&nbsp;The only reason is that the methods in DELG can not be reproduced in  "+1M" dataset setups when we use the source code published by the author. The computing process would approximately take more than 690 days. We checked it again and again. If you don't believe us, you can try reproducing it yourself.
>
> Q4. "In Tab 2, the authors simply take latency numbers reported in previous work and compare to their own for the new method, using a different GPU. The comparison does not seem appropriate."
>
> A4: As illustrated in the caption of tab2, UGALR* refers to a version of UGALR with the same extraction settings as Group 3.  In the comparison in terms of time and memory, all the models were tested on the same NVIDIA Tesla P40 GPU, with images resized into 1024. To ensure the fairness of the experiment, only models were different and we kept the other settings the same.  Consequently, the experiment is rigorous.
>
> &nbsp;&nbsp;&nbsp;&nbsp;Thanks again for your careful readings and patience. We will appreciate it if you could give us a chance to see the innovative ideas in this paper. They are significant. It uses CNN instead of RANSAC to learn homography transformation and introduce intermediate supervision, which is not studied in image retrieval yet.

---

> > ### Comment · Reviewer_wJ79 · 2021-11-25
> > **Response to author's Q2-3-4 rebuttal**
> >
> > Q2. Thank you for acknowledging and improving upon the errors.
> >
> > Q3. Just because the authors are not able to reproduce the results, it does not mean that they are not correct. At the very least, if the authors are suspicious or have difficulties reproducing results, this should be stated in the submitted paper manuscript, instead of simply omitting the results. Furthermore, the authors' reasoning is unfortunately absolutely incorrect. One does not need to do local feature matching for all images in the R1M index. Local feature matching only needs to be done for the top 100 retrieved images for each query. Since there are 70 queries, only 7000 pairwise matching operations are needed. If the number of "30 images takes half an hour" is correct, then 7000 would take 4.86 days, completely different from the 695 days mentioned in the rebuttal by the authors.
> >
> > "Our reproducing process is based on the source code published by the authors of DELG on github and we checked it again and again. It is really weird."
> > -> Unfortunately, you'd need to check again. The DELG code clearly only re-ranks 100 images at most per query.
> >
> > Q4. Thanks for the clarification.

---

### Official Review · Reviewer_UMH7 · 2021-11-02

**Correctness:** 3
**Technical Novelty And Significance:** 3
**Empirical Novelty And Significance:** 2
**Recommendation:** 6
**Confidence:** 3

**Main Review:**

Strengths:
* the proposed architecture is deeply described, justifying each of its parts;
* the experiments are performed using two datasets and the proposed model is compared with a lot of other models, providing a serious and robust discussion;
* the ablation analysis is complete and justifies the use of each component of the architecture, describing numerically why each macro-layer (e.g. LALM) helps to improve general performances;
* all the results against the SOTA models are complete and well described;
* one of the main (well-described) goals of the architecture is to speed up the retrieval process reducing the used space. A comparison of these aspects has been clearly done, demonstrating the great advantages of UGALR with respect to the other models.
* the benefits in terms of time and space do not preclude an improvement in accuracy performance, which is higher than in the baseline models.

Weaknesses:
* even if complete, the description of the architecture seems a little bit confusing. For example, the use of capital letters in figure 1 (and then in the text)  is confusing;
* the comparison in terms of time and memory seems ok but is not very clear if all the models were tested on the same GPU (using the same environment). If not, this is a real problem and the results cannot be taken into account;
* there is no error analysis. The accuracy is higher with respect to baseline models, but not that much; in order to study why, an error analysis would be helpful, maybe providing a comparison with the common errors of the other SOTA models.

Minor grammar errors:
* in figure 2 you use $\varphi$ (varphi) but in the text $\phi$ (phi) is always used;
* lack some spaces between words and brackets (but this could be a template prerogative)


**Summary Of The Paper:**

This paper proposes a new pipeline architecture for image retrieval. The main purpose of the proposed architecture is to overcome the time and space consumption due to the images' re-ranking that exploits local features (after the search using global features).
In order to do that, UGALR is proposed as a single-stage pipeline able to combine global and local features to speed up the retrieval process, removing the re-ranking step. This is achieved using a CNN able to learn the homography transformation in local feature matching.
Moreover, the architecture combines spatial and channel attention with the aid of intermediate supervision, obtaining better performances than the SOTA models presented in the analysis of the results. This is achieved with the introduction of a Location Attention Learning Module (called LALM).


**Summary Of The Review:**

The paper describes the proposed architecture proving a complete explanation of all of its parts (with the help of a complete ablation analysis). The comparison with a lot SOTA models on two datasets confirms the robustness of the proposed model. Even if the improvements in terms of accuracy are not so high (not a negative point, there is however a clear improvement) there is a huge improvement in terms of speed and memory used. An improvement for the paper would be the addition of a complete error analysis that now is not present and a better description of the entire architecture that, at present, is not very clear. Moreover, is not clearly specified if all the models have been tested on the same GPU (by the authors) to test speed and memory usage, this is an important point to specify because is one of the main goals of the paper.

---

> ### Author Response · Authors · 2021-11-17
> **Response to Reviewer UMH7**
>
> Thanks for your careful readings and valuable comments about our papers, and we will address some concerns below.
>
> Q1. presentation and grammar errors
>
> A1:  We should apologize for our mistakes. In the latest draft, we added the explanation of capital letters in the caption of figure 1. And we have fixed the inconsistency between  $\phi$  in figure 2 and $\varphi$ in the text. Thanks very much.
>
> Q2. "the comparison in terms of time and memory seems ok but is not very clear if all the models were tested on the same GPU (using the same environment). If not, this is a real problem and the results cannot be taken into account."
>
> A2: In the comparison in terms of time and memory, all the models were tested on the same NVIDIA Tesla P40 GPU, with images resized into 1024. To ensure the fairness of the experiment, only models were different  and we kept the other settings the same.  Consequently, the experiment is rigorous.
>
> Q3. “There is no error analysis. The accuracy is higher with respect to baseline models, but not that much; in order to study why, an error analysis would be helpful, maybe providing a comparison with the common errors of the other SOTA models.”
>
> A3:  Thanks for your suggestions.  Experiment 1 is to prove that UGALR could achieve state-of-the-art performance with less memory and less extraction latency compared with other approaches. And UGALR focuses on reducing memory and extraction latency. It's enough that the accuracy is slightly higher with respect to baseline models. It's a surprising result that an end-to-end approach could slightly outperform traditional two-stage retrieval methods with less memory and less extraction latency. The reason why we didn't make an error analysis is that it rarely appears in the image retrieval field. But we think your comments are really valuable. And if necessary, we will add this experiment in the next version of our draft.
>
> &nbsp;&nbsp;&nbsp;&nbsp;Thanks again for your careful readings and valuable comments about our papers, and we've learnt a lot from your comments.  We would further improve our draft if you could give us a chance. We hope using CNN instead of RANSAC to learn homography transformation and introducing intermediate supervision into image retrieval would be significant topics in the future.

---

> > ### Comment · Reviewer_UMH7 · 2021-11-29
> > **Feedback confirmation**
> >
> > The fact that the same GPU has been used for all the experiments is good. Moreover, the other answers are good enough to confirm my review and my recommendation rating.

---

### Official Review · Reviewer_N1bK · 2021-11-02

**Correctness:** 2
**Technical Novelty And Significance:** 2
**Empirical Novelty And Significance:** 2
**Recommendation:** 3
**Confidence:** 5

**Details Of Ethics Concerns:**

1. Main contributions are not clear. The main structure is simple and normal, where the attention techniques are adopted to extract the local information.
2. I do not think re-ranking is an indispensable part of an image retrieval system. Thus, the motivation of this manuscript is not valid enough.
3. Section 3 is described as shallow, which prevents readers from understanding your main idea.
4. The influence of free parameters should be studied.
5. More latest image retrieval models should be added to testify your model.


**Main Review:**

The proposed method is feasible. However, its novelty is pretty limited. Also, many points are confusing for readers.

**Summary Of The Paper:**

In this work, the authors propose a Unifying Global and Attention-based Local Features Retrieval method (referred to as UGALR). UGALR accelerates extraction speed and reduces memory consumption by removing the re-ranking process and learning local feature matching with convolutional neural networks instead of the RANSAC algorithm. In addition, UGALR learns more accurate and semantic local information by combining spatial and channel attention with intermediate supervision.

**Summary Of The Review:**

Due to the limited contributions, I do not recommend this paper to publish on ICRL.

---

> ### Author Response · Authors · 2021-11-17
> **Response to Reviewer N1bK**
>
> Q3. "The influence of free parameters should be studied."
>
> A3: We studied the influence of free parameters before, but the total length of the paper is over 9 pages, so we deleted it. Indeed, it is important, so we decided to add the related experiment to the appendix in the next draft.
>
> Q4. "More latest image retrieval models should be added to testify your model."
>
> A4: As mentioned in Introduction, the state-of-the-art method in image retrieval is DELG[1]. Consequently, we lay more emphasis on comparing our UGALR with DELG, and if ours outperforms DELG, it means UGALR has achieved state-of-the-art performance because DELG has been compared with numerous image retrieval models in [1]. In addition to DELG, we also compared with two local features-based methods: HesAff-HardNet[2] and DELF[3] and more than five global features-based methods,such as SOLAR[4], R101-GEM[5] and R101-R-MAC[6]. In total, 8 approaches were compared with our UGALR. Experiments show that UGALR outperform others greatly in accuracy with less memory and less extraction latency, which has achieved state-of-the-art performance. But if you think they are not enough, we would add more latest image retrieval model to testify our model in the next version of our draft.
>
> Reference:
> 1. Bingyi Cao, André Araujo, Jack Sim: Unifying Deep Local and Global Features for Image Search. ECCV (20) 2020: 726-743
> 2. Dmytro Mishkin, Filip Radenovic, Jiri Matas: Repeatability Is Not Enough: Learning Affine Regions via Discriminability. ECCV (9) 2018: 287-304
> 3. Hyeonwoo Noh, Andre Araujo, Jack Sim, Tobias Weyand, Bohyung Han: Large-Scale Image Retrieval with Attentive Deep Local Features. ICCV 2017: 3476-3485
> 4. Tony Ng, Vassileios Balntas, Yurun Tian, Krystian Mikolajczyk: SOLAR: Second-Order Loss and Attention for Image Retrieval. ECCV (25) 2020: 253-270
> 5. Filip Radenovic, Giorgos Tolias, Ondrej Chum: Fine-Tuning CNN Image Retrieval with No Human Annotation. IEEE Trans. Pattern Anal. Mach. Intell. 41(7): 1655-1668 (2019)
> 6. Albert Gordo, Jon Almazán, Jérôme Revaud, Diane Larlus: End-to-End Learning of Deep Visual Representations for Image Retrieval. Int. J. Comput. Vis. 124(2): 237-254 (2017)
>
> &nbsp;&nbsp;&nbsp;&nbsp;Thanks for your suggestions. We try to address your concern above, and we will further improve our draft in the final version if you could give me a chance to see the innovative ideas in this paper. They are significant. It uses CNN instead of RANSAC to learn homography transformation and introduces intermediate supervision, which are not studied in image retrieval yet. Experiments in ablation study also verified their effectiveness.

---

> > ### Comment · Reviewer_N1bK · 2021-11-25
> > **Response to author's rebuttal**
> >
> > Although the authors have been modified some issues, the novelty of this manuscript is low that cannot meet the demands of ICLR.
> > I still insist on my original decision.

---

> ### Author Response · Authors · 2021-11-17
> **Response to Reviewer N1bK**
>
> Thanks for your comments about our papers.
>
> &nbsp;&nbsp;&nbsp;&nbsp;To address your concern, we’d like to emphasize our contributions. The primary contribution is to learn homography transformation with CNN and replace the re-ranking process with information fusion to obtain more powerful features, which overcomes the low efficiency of local features in storage and matching. Different from traditional global and local features-based approaches that rank twice, ours only needs ranking once with less memory and time to achieve the sota results. In addition, we introduced intermediate supervision into image retrieval for the first time, which differentiates the importance of features and attention to avoid attention degradation. The two innovations are not studied by previous works yet and would be very meaningful research directions in the future. Furthermore, we also explored enhancing local attention in both channel and spatial dimensions.
>
> &nbsp;&nbsp;&nbsp;&nbsp;Next, we'd like to answer some of your questions.
>
> Q1. "I do not think re-ranking is an indispensable part of an image retrieval system. Thus, the motivation of this manuscript is not valid enough."
>
> A1:  If we pursue better retrieval performance, using local features to re-rank is of great significance because global features-based approaches lack spatial arrangement information and local features that contain geometric information could make up with it. And numerous approaches have proved that methods with local features re-ranking greatly outperforms those without it[1][2][3].
> &nbsp;&nbsp;&nbsp;&nbsp;Although approaches that first search with global features then re-rank with local feature matching could achieve state-of-the-art retrieval performance, they have two big disadvantages. The first disadvantage is that the processes of extracting local features, local feature matching, and re-ranking are of low efficiency either in memory or in latency. The second disadvantage is that  the local feature learning in these methods is not accurate enough and semantic enough because of the trivial design.
> &nbsp;&nbsp;&nbsp;&nbsp;In this work, we proposed a Unifying Global and Attention-based Local Features Retrieval method (referred to as UGALR), which is an end-to-end and single-stage pipeline. It benefits from two aspects: 1) it accelerates extraction speed and reduces memory consumption by removing the re-ranking process and learning local feature matching with convolutional neural networks instead of RANSAC algorithm; 2) it learns more accurate and semantic local information through combining spatial and channel attention with the aid of intermediate supervision. Experiments on Revisited Oxford and Paris datasets validate the effectiveness of our approach, and we achieved state-of-the-art performance compared to other popular methods  with lower memory occupation and faster extraction speed.
> &nbsp;&nbsp;&nbsp;&nbsp;These are the background and our motivation.
>
> Reference:
> 1. Bingyi Cao, André Araujo, Jack Sim: Unifying Deep Local and Global Features for Image Search. ECCV (20) 2020: 726-743
> 2. Paul-Edouard Sarlin, Cesar Cadena, Roland Siegwart, Marcin Dymczyk: From Coarse to Fine: Robust Hierarchical Localization at Large Scale. CVPR 2019: 12716-12725
> 3. Hajime Taira, Masatoshi Okutomi, Torsten Sattler, Mircea Cimpoi, Marc Pollefeys, Josef Sivic, Tomás Pajdla, Akihiko Torii: InLoc: Indoor Visual Localization with Dense Matching and View Synthesis. IEEE Trans. Pattern Anal. Mach. Intell. 43(4): 1293-1307 (2021)
>
> Q2. "Section 3 is described as shallow, which prevents readers from understanding your main idea."
>
> A2: The main idea of section 3 contains three steps. First, we enhance local attention learning in two dimensions: spatial and channel, to learn more accurate and semantic attention information. Second, we apply intermediate supervision in training local attention modules to adaptively learn the local key point information about category responses. And intermediate supervision helps avoid attention degradation. Third, we concatenate the output of local attention module with the blocks on the backbone, and the global branch takes into the local key point information to  learn  homography transformation to boost global feature learning.
> &nbsp;&nbsp;&nbsp;&nbsp;In the latest draft, we made our description deeper and made it easier to understand. We hope that you could give us another chance to see the innovation of this paper.

---

### Official Review · Reviewer_MPBM · 2021-11-02

**Correctness:** 3
**Technical Novelty And Significance:** 3
**Empirical Novelty And Significance:** 2
**Recommendation:** 5
**Confidence:** 5

**Main Review:**

Strengths:
- The paper proposes an interesting approach that obtains very good results using only global features, outperforming two-stage methods that use local features for re-ranking.
- Experimental evaluation is also complete and thorough, with a very extensive ablative study.

-----
Weaknesses:
- The main weakness of the paper it has to do with its presentation: it has numerous syntax and grammatical errors and typos, which makes for a challenging read sometimes. Some paragraphs are difficult to follow and in my opinion should be rewritten. The main problem with this is that it affects the motivation of the paper: I feel I couldn't fully grasp the rationale behind some design choices or the conclusions drawn from some of the experiments. It is difficult to provide a complete list of all the things that should be addressed by the authors but I'll discuss in section "Presentation" below what I think are the most importante ones. Nevertheless, I highly encourage the authors to do another pass on the writing.
- Slightly related to the point above, I'm missing a deeper and more detailed explanation about the claim that the last convolutional blocks act as an homography transformation because there's no experimental or theoretical study that proves that in the text. Overall, the contributions of the paper are not clear enough.
- Not too important: I understand that the focus of the paper is removing the re-ranking step, but it would have been interesting to see if the local features produced by the LALM module could have been used for this purpose to further improve the results in applications that can afford the extra memory and latency costs.
- I wondered why the authors decided to use average-pooling and cross-entropy loss for the intermediate supervision instead of GeM or ArcFace as for the global features. Any insight about why this shouldn't work better?
- Apologies if I missed this in the text, but how are R50-UGALR* and R50-UGALR in Table 2 are different? I couldn't find a reference in the text to the dimensionality reduction mentioned here.

Presentation:
- The caption of Figure 1 only half describes the diagram there so I would suggest to extend it to give a better overview of the method. It is also missing the legend/explanation of what is "A", "X", "M", and "C".
- "At present, there is such a problem in image retrieval: although the results of global feature-based retrieval are correct in the category, they are not sorted in detail" -> technically, the problem in instance-retrieval datasets is that they're not correct in the category so that's why re-ranking is performed.
- "But if we don’t differentiate the importance of attention and features, it’s hard to make sure that the learned attention is not part of features, which is not an ideal way for attention learning." -> this sentence should be re-written
- "However, these methods have not been well modified yet to solve image retrieval problems." -> this sentence should be re-written
- "in the next small section." -> in the next section
- The nomenclature used in Figure 2 does not match with the text nor with the formulas in Eq 1-4
- In page 4 the text mentions "D", which I'm assuming it's the output of "X" in Figure 1, but it has never been defined before.
- The previous to last paragraph in page 4 is difficult to follow and it should be re-written more clearly. This is a crucial paragraph to properly understand the rationale behind the LALM module and the intermediate supervision and in its current shapes it loses impact.
- "We assume that there is an identical average target available for every category of objects" -> I couldn't understand this assumption
- Eq 6 makes reference of La but it was never defined in the text before (only in figure 1)
- "Obviously, UGALR completely outperforms previous global features based methods" -> "As we can see, UGALR..."
- "The most amazing two improvements we made are" -> "The largest two improvements..."
- "Amazingly, our approach using only global features outperforms" -> "Surprisingly, our approach..."
- Spaces are usually missing just before parenthesis

**Summary Of The Paper:**

This paper addresses the problem of image retrieval and proposes a method that tries to learn better representations through the use of local feature attention and reduces memory and latency overheads by not requiring re-ranking with local features and only using global one.

For this, the paper proposes a module called Local Attention Learning Module (LALM) that performs both spatial- and channel-wise attention on local features and that it's plugged between two consecutive blocks of the backbone architecture. The output of this module is then concatenated to the input of the next block. The rationale behind this is that the following blocks will act as an homography transformation of the local features and will produce a better global representation. The method also uses intermediate supervision by computing a cross-entropy loss on top of the output features produced by the LAL Module.

**Summary Of The Review:**

Overall, even though it doesn't introduce any impactful contribution, I think that the paper proposes an interesting method that is able to obtain very good results using only global features, outperforming methods that also combine re-ranking with local features. However, I find it difficult to recommend this paper for its acceptance in its current shape since its presentation is not adequate. Introduction and Method sections are sometimes difficult to follow and I couldn't fully grasp some of the ideas and rationale behind the design choices. Another pass in the text is required, and figures and formulas should also be updated and improved.

---

> ### Author Response · Authors · 2021-11-16
> **Response to Reviewer MPBM**
>
>
>
> Thanks for  your careful readings and valuable comments about our papers, and we will address some concerns below.
>
>
> Q1. the problems of presentation
>
> A1：Thanks for your valuable comments. Since the first version of this paper is more than 9 pages long, in the process of reducing length, we made some errors and accidentally missed some important content. And we should apologize for that. In the latest draft, we have fixed the errors you mentioned in "Presentation". Thanks very much.
>
> Q2. "We assume that there is an identical average target available for every category of objects -> I couldn't understand this assumption"
>
> A2：In instance-level image retrieval, images in one category describe the same object. So we could assume that there is an identical average target available for each category of objects. Then, from each image in one category to its corresponding target image exists a homography matrix. For example, if we take the front image of the object as the standard target, then each image of this category could be transformed to the standard target using a homography matrix. The mission of our network is to learn homography transformation between them.
>
> Q3. "Slightly related to the point above, I'm missing a deeper and more detailed explanation about the claim that the last convolutional blocks act as an homography transformation because there's no experimental or theoretical study that proves that in the text. Overall, the contributions of the paper are not clear enough."
>
> A3: First, we would give a deeper and more detailed explanation about the claim that the last convolutional blocks act as a homography transformation. Second, we will talk about the experimental parts of homography in this paper.
>
> First, according to this paper, the LALM module outputs feature maps that contain local key point information (attention is to find key point response and  softplus activation act as a threshold filter to extract local key point information). Then we concatenated it with the global branch, and the subsequent blocks would take into these local key point information to learn homography transformation illustrated above. The reason why the block could learn homography transformation is that homography transformation is linear transformation. And linear transformation could be learnt by CNN.
>
> Second, the experimental study of homography transformation is in table 4 and its corresponding Local Attention Learning Module part. The  first  row  denotes the  performance  of  not  using  the  LALM  block  and  the  second  row  denotes  the  performance  of using  LALM.  Not using LALM means there is a single global branch in the network and means not learning homography transformation. According to this experiment, the  model  with  LALM  outperforms  the  model  without  it  by  1.77%  inROxf-M,1.27% inROxf-H, 1.85% inRPar-M and 3.74% inRPar-H. As we can see, the homography transformation learning with LALM could boost retrieval performance.
>
> Q4. "- I wondered why the authors decided to use average-pooling and cross-entropy loss for the intermediate supervision instead of GeM or ArcFace as for the global features. Any insight about why this shouldn't work better?"
>
> A4: The aim of intermediate supervision is to help learn attention that contains local key point information, and these key point information is essentially the response of the category. In this background, using arcface loss in intermediate supervision would lead to insensibility of the category response since arcface loss has more difficulty in convergence compared with cross entropy loss. In addition, compared with avg-pooling, GEM would amplify category response. We hope that the key point information the model learned is also obvious and effective even if there is no amplification of GEM. Consequently, we chose avg-pooling and cross entropy loss here.
>
> Q5. "- Apologies if I missed this in the text, but how are R50-UGALR* and R50-UGALR in Table 2 are different? I couldn't find a reference in the text to the dimensionality reduction mentioned here."
>
> A5: As described in the caption of table 2, UGALR* refers to a version of UGALR with the same extraction settings with Group 3. In the latest draft,  we also emphasized it in the text.
>
> Q6. "- Not too important: I understand that the focus of the paper is removing the re-ranking step, but it would have been interesting to see if the local features produced by the LALM module could have been used for this purpose to further improve the results in applications that can afford the extra memory and latency costs."
>
> A6: Thanks, and this network has used the local features produced by LALM to further improve the results through concatenation as illustrated in A2 and A3.
>
> Thanks for your valuable suggestions. We have fixed some errors that you mentioned, and we will further improve our draft in the final version if you could give me a chance to see the innovative ideas in this paper.

---

### Decision · Program_Chairs · 2022-01-20

**Decision:**

Reject

**Comment:**

The paper proposes a new method to combine global and local image features, targeted at image retrieval applications. The main idea is a model branch where both spatial and channel attention are used. The local feature branch undergoes supervision directly and this branch’s output is concatenated to the global feature branch’s output in order to eventually produce the final image embedding.

The reviewers appreciated the care in the evaluation (ablative analysis) and the promise of the approach compared to existing baselines. The reviewers also expressed concerns about several claims, for instance that the proposed approach is able to learn homography transformations, the quality of the exposition, and missing baselines. The reviewers also pointed out that several parts of the paper were hard to follow and important details were missing.

The authors submitted responses to the reviewers' comments. After reading the response, updating the reviews, and discussion, the reviewers considered that ‘the concatenation of local features with global ones works does not mean at all that some geometric transformation is learned’ and the justification provided for omitting baselines (suggested by the reviewers) were unconvincing. The feedback provided was already fruitful, yet major issues still remain.

We encourage the paper to pursue their approach further taking into account the reviewers' comments, encouragements, and suggestions. The detailed feedback lays out a clear path to generate a stronger submission to a future venue.

Reject.